# Tuning CAR-T cells by targeting cancer-associated glycan in pancreatic cancer

Sangwoo Park [1,2,3], Cassidy E. Ho[1,2], Eli P. Darnell[1,2,3], Alexandra N. Wolff[1,2,3], Hana Takei [1,2], Filippo Birocchi [1,2,3], Amanda A. Bouffard[1,2], Diego Salas-Benito[1,2,3], Giulia Escobar [1,2,3], Mark B. Leick [1,2,3], Adele Mucci[1,2], Trisha R. Berger[1,2] & Marcela V. Maus [1,2,3] ✉

Chimeric antigen receptor (CAR) T cell therapy has transformed cancer treatment but its efficacy remains limited in solid tumors due to antigen heterogeneity, an immunosuppressive microenvironment, and the glycocalyx barrier. The glycocalyx, composed of dense glycoproteins such as MUC1, is markedly expanded in cancers, where it impedes immune cell access and antigen engagement, thereby reducing efficacy. In most adenocarcinomas, Tn antigen, comprising N-acetylgalactosamine linked to serine or threonine, is overexpressed. Tn-MUC1, a truncated form of MUC1 decorated with Tn antigen, is frequently overexpressed in pancreatic cancer. Here, we incorporate a non-signaling glyco-bridge binder recognizing Tn-MUC1 into mesothelin-directed CAR-T cells. This bridge enhances tumor recognition and cytotoxicity by increasing avidity and facilitating CAR activation in a density- and affinity-dependent manner. To broaden its applicability, we design a tandem Helix pomatia agglutinin (HPA) lectin-based bridge that recognizes Tn antigens across cancer types. CAR-T cells with the HPA-bridge exhibit superior cytotoxicity in pancreatic cancer models.

Chimeric antigen receptor T (CAR-T) cell therapy has revolutionized cancer immunotherapy by harnessing the body's immune system to target and eliminate malignant cells. It has dramatically changed the treatment landscape for patients with relapsed and refractory B-cell malignancies, especially following FDA approval for diffuse large B-cell lymphoma, B-cell acute lymphoblastic leukemia, and multiple myeloma[1–5]. This success has sparked significant interest in expanding the technology to solid tumors. CAR-T cell studies targeting various solid cancers, including pancreatic cancer and triple-negative breast cancer, have recently gained momentum, although challenges remain that can hinder CAR-T cell efficacy and persistence[6–11]. These challenges include antigen heterogeneity, an immunosuppressive tumor microenvironment, and physical barriers such as the tumor stroma and cellular glycocalyx[12–17].

The physical barriers in solid tumors present significant challenges to immune cells, including CAR-T cell therapy. In particular, the

dense layer of heavily glycosylated proteins like mucins, which are often overexpressed in cancers, can modulate the tumor microenvironment that hinders immune cell infiltration and effective immune cell targeting[8,17–22]. The cancer-associated mucin MUC1, decorated with N-acetylgalactosamine (GalNAc) linked to serine or threonine residues (known as the Tn antigen; Tn-MUC1), is frequently found in solid tumors such as pancreatic cancer and has become a target for several therapeutic strategies, including CAR-T cell therapy[23–31]. Specifically, CAR-T cells targeting Tn-MUC1 developed by Posey et al. were advanced to a clinical trial with patients with Tn-MUC1-overexpressing cancers, including pancreatic cancer and triple-negative breast cancer, and no significant safety concerns or on-target/off-tumor toxicity were reported (NCT04025216)[23,32]. However, glycoproteins and other large components of the glycocalyx are thought to sterically shield molecular epitopes, affecting the interactions between immune cells and tumor cells[12]. Aberrant N-glycosylation,

[1]Krantz Family Center for Cancer Research, Massachusetts General Hospital, Boston, MA, USA. [2]Cellular Immunotherapy Program, Massachusetts General Hospital Cancer Center, Boston, MA, USA. [3]Harvard Medical School, Boston, MA, USA. ✉e-mail: mvmaus@mgh.harvard.edu

such as β1,6-GlcNAc-branched N-glycans synthesized by MGAT5, has been shown to promote immune evasion by masking tumor-associated antigens and contributing to T cell dysfunction and exhaustion[33]. Overexpression of MUC1 contributes to a thickened glycocalyx, preventing access to key receptors and/or antigens needed for immune cell killing[12,17,20]. Overcoming this barrier by designing CAR-T cells that can bind the glycocalyx effectively may significantly enhance their ability to bind their target antigen and subsequently induce tumor cell death, thereby improving therapeutic efficacy in solid cancers, like pancreatic cancer.

Glycans, which are sugar molecules attached to cell-surface proteins, can alter the functions of essential immune cells, including macrophages, dendritic cells, cytotoxic T cells, and natural killer cells[34]. The Tn antigen is a carbohydrate antigen overexpressed in various cancers, including pancreatic, breast, ovarian, bladder, prostate, lung, and stomach cancers[35–41]. It is primarily overexpressed when T-synthase, the enzyme that transfers galactose from UDP-Gal to Tn antigen to form Core 1 O-glycans, is degraded, silenced, or mutated. These alterations often occur due to defects in Core 1 β3-Gal-T-specific molecular chaperone (*COSMC*), which prevents protein misfolding of T-synthase[42]. Loss of functional *COSMC* through mutation, deletion, or hypermethylation can also contribute to Tn antigen expression in some cancer cell lines and malignancies[43–46]. For example, the Jurkat cell line overexpresses the Tn antigen due to a loss-of-function mutation in *COSMC*[43]. Additionally, the initiating enzymes, polypeptide N-acetylgalactosaminyltransferase (*ppGalNAc-Ts*), transfer GalNAc to serine and threonine residues of proteins. Several isoforms of *ppGalNAc-Ts*, including T1, T3, T6, and T13, are elevated in human cancers, resulting in increased Tn antigen expression[47–50]. For instance, overexpression of GalNAc-transferase 3 (*GalNAc-T3*) has been shown in pancreatic cancer and has been linked to promoting cancer cell growth[51]. Due to this characteristic overexpression pattern, the Tn antigen serves as a potential biomarker for early prediction or detection of human cancers[52,53].

Here, we engineer CAR-T cells to express non-signaling glycocalyx-binding molecules, termed the glyco-bridge. This design strategically utilizes cancer-associated MUC1 to enhance binding efficiency, thereby improving CAR-T cell activity against mucin-overexpressing cancer cells. This approach leverages the unique structural characteristics of the glycocalyx to facilitate more effective targeting of the cancer cell by increasing adhesion of the CAR T cell. Although not originally intended, we found that the glyco-bridge alone could unexpectedly modulate CAR-T cell activation, depending on the target antigen density on cancer cells. To target a broader range of cancer-associated Tn antigens beyond Tn-MUC1, we incorporated tandem Helix pomatia agglutinin (HPA) lectins as the antigen-binding domain of the CAR. These tandem HPA-based CAR-T cells show enhanced efficacy compared to single HPA-based CAR-T cells against cancer cell lines and patient-derived xenograft (PDX) models of pancreatic cancer in vitro and in vivo. However, HPA-based CAR-T cells exhibited toxicity due to on-target/off-tumor binding. To mitigate this, we next incorporate HPA lectins into the glyco-bridge approach to enable selective targeting of universal Tn antigens on cancer cells. Our results reveal that glycan-targeting strategies using a bridge system that binds to cancer-associated glycans enhance CAR-T cell-mediated cytotoxicity.

## Results
### Glyco-bridge strategy enhances CAR-T cell killing of Tn-MUC1-expressing cancer cells

Tn antigens are overexpressed due to the upregulation of *ppGalNAc-Ts*, genetic silencing or mutation of *C1GALT1*, or mutations in *COSMC*[43–50] (Fig. 1a). Tn antigens decorate the backbone of cell-surface glycolipids and glycoproteins, such as cell-surface mucins. Notably, cancer-associated mucins are overexpressed in tumors and cell lines and are associated with high heterogeneity at the cell surface level[54,55].

Since the mucins can form a nanoscale glycocalyx barrier at the cellular interface that impedes immune cell interaction, we tested the hypothesis that enhancing CAR T cell binding to cancer cells could help overcome this barrier. Building on previous approaches that utilized two CAR constructs[56–58], we designed our CAR construct with a non-signaling bridge system. In contrast to prior methods, we employed a distinct transmembrane domain to prevent hetero-dimerization and removed the intracellular domain from the bridge construct. We used this modification with the hypothesis that the bridge would overcome the glycocalyx barrier solely by facilitating binding of the CAR T cell to the cancer cell (Fig. 1b). Our glyco-bridge was designed to engage the cancer-associated, cell surface mucin structure Tn-MUC1 with a single-chain variable fragment (scFv). We initially tested a Tn-MUC1 scFv (5E5)[23] as the glyco-bridge binder alongside a mesothelin-targeting CAR molecule (with an SS1 scFv)[59] and used a CD19 scFv (FMC63)[60] bridge as a control, since CD19 is not expressed by solid tumor cells. All of these scFvs have been used in CAR T cells that have either entered clinical trials or have received FDA approval[2,23] and contain a $(G_4S)_3$ linker between their variable heavy and light chains. For the CAR construct, the SS1 scFv was directly fused to a CD8 hinge/transmembrane domain, a 4-1BB costimulatory domain, and CD3ζ (Fig. 1c). The bridge constructs incorporated a CD28 hinge/transmembrane domain to prevent dimerization with the CAR molecule. To better anchor the bridge to the cell membrane, we incorporated CD28 intracellular domain that we mutated in its key subdomains to prevent signaling. Specifically, we mutated proline residues P208 and P211 to alanine within the PRRP subdomain to prevent SH3-containing proteins (such as Itk and Tec) from binding, and we modified the PYAP motif by substituting tyrosine (Y191) with phenylalanine, inhibiting binding to the SH2 domain of Lck, a key mediator in T-cell signaling[61] (Fig. 1b). Expression of the CAR and glyco-bridge molecules was confirmed using an ALFA tag and the $(G_4S)_3$ linker (Fig. 1d). In both constructs, the transduction rate of CAR molecules ranged from 40 to 50% and remained stable over time. The CD19-bridge alongside the CAR molecule demonstrated strong binding affinity to recombinant CD19 protein (Supplementary Fig. 1a).

We used the Capan-2 pancreatic ductal adenocarcinoma (PDAC) cell line, which expresses high levels of cell-surface mucins, including MUC1 (Supplementary Fig. 1b), as target tumor cells. This cell line has previously been used in studies evaluating 5E5 CAR-T cells targeting Tn-MUC1, demonstrating functional recognition and cytotoxicity[23]. They also express high levels of mesothelin (the target of the CAR) and lack CD19 expression (the target of the control bridge). To further increase the surface expression of Tn-MUC1 for proof-of-concept evaluation of the glyco-bridge strategy, we generated *C1GALT1* knockout Capan-2 cells using CRISPR/Cas9-mediated gene editing or treated the cells with itraconazole, an antifungal drug, to inhibit C1GALT1 enzymatic activity[62] (Fig. 1e). Both Capan-2 *C1GALT1* KO cells and itraconazole-treated Capan-2 cells had increased Tn-MUC1 expression. In a real-time killing assay with Capan-2 *C1GALT1* KO cells, mesothelin-targeted CAR-T cells incorporating a Tn-MUC1-bridge demonstrated higher killing efficiency compared to those with the control CD19-bridge (Fig. 1f, g and Supplementary Fig. 1c). Likewise, combining itraconazole treatment with Tn-MUC1 bridge CAR-T cells further enhanced killing of Capan-2 cells in vitro (Fig. 1h, i). To determine if this effect would be applicable to either bridge molecule, we overexpressed CD19 in Capan-2 cells. The CD19-bridge enhanced the cytotoxicity of mesothelin-targeted CAR-T cells. To test this concept further, we utilized the RPMI 8226 multiple myeloma cell line, which also exhibits high expression of MUC1 but lacks detectable mesothelin and CD19. After CD19 overexpression, we observed a similar enhancement in BCMA-targeted CAR-T cell cytotoxicity, suggesting that the bridge strategy is compatible with different CAR constructs and tumor types. (Supplementary Fig. 1d–g).

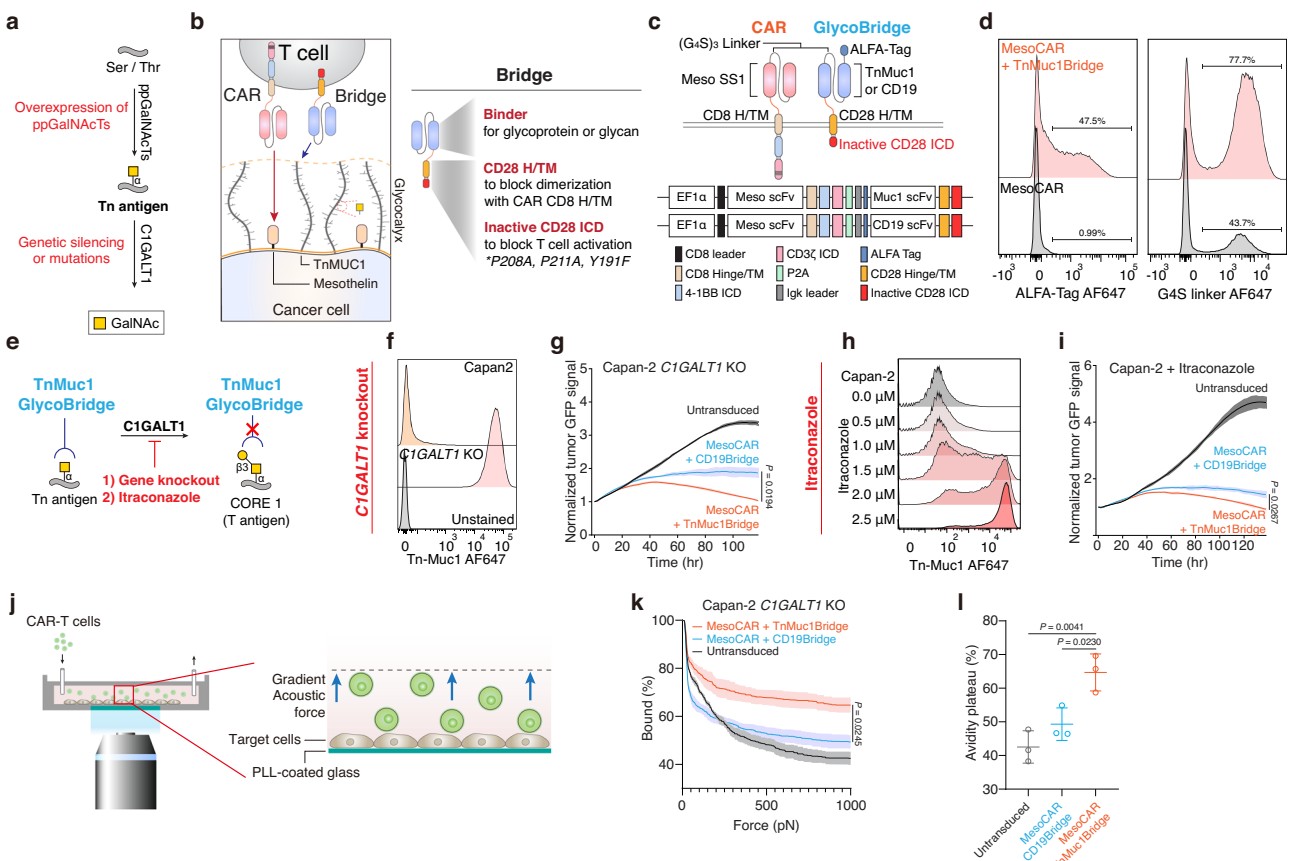

**Fig. 1 | Enhancing CAR-T cell efficacy by inducing Tn antigen expression.**
**a** Biosynthetic pathway for O-glycans with polypeptide
N-acetylgalactosaminyltransferases (*ppGalNAcTs*) and *C1GALT1* gene. **b** Schematic
of mesothelin-targeting Chimeric Antigen Receptor (CAR)-T cells with glyco-
bridge. **c** Graphic representation of the constructs used to make CAR-T cells with
Tn-MUC1 bridge or CD19-bridge. The glyco-bridge contains a CD28 hinge and
transmembrane domain (H/TM) and an inactive CD28 intracellular signaling
domain (ICD). **d** Flow cytometry analysis of glyco-bridge surface levels on T cells
using ALFA-Tag (left) and $(G_4S)_3$ linkers expression levels (right). **e** Schematic of the
biosynthetic pathway for Tn antigen. **f** Flow cytometry analysis of Tn-MUC1 surface
level on Capan-2 and Capan-2 *C1GALT1* KO cells. **g** Representative real-time cyto-
toxicity assay against Capan-2 *C1GALT1* KO cells at a 1:1 E:T ratio (relative to day 0
tumor seeding) from $n = 3$ distinct human blood cell donors. Results are mean ± s.d.
of $n = 3$ independent measurements. **h** Surface level of Tn-MUC1 on Capan-2 fol-
lowing itraconazole treatment at the indicated concentrations for 48 h.

**i** Representative real-time cytotoxicity assay against 2.5 μM itraconazole-treated
Capan-2 cells at a 1:1 E:T ratio (relative to day 0 tumor seeding) from $n = 3$ three
distinct human blood cell donors. Results are mean ± s.d. of $n = 3$ independent
measurements. **j** Schematic of cell avidity measurement with acoustic force
microscopy. **k** Strength of interaction between Capan2 *C1GALT1* KO target cells and
CAR-T cells expressing TnMUC1 or CD19-bridges. Percentage of total CAR-T cells
remaining bound to target cells as the acoustic force ramp is applied from 0 to
1000 pN are shown. Results are mean ± s.e.m. of $n = 3$ independent measurements.
**l** Percentage of CAR-T cells remaining bound to the target cells at the avidity
plateau under 1000 pN force from Fig. 1k. Results are mean ± s.d. of $n = 3$ inde-
pendent measurements. In **d**, **f**, **h**, representative flow cytometry data from three
independent experiments. In **g**, **i**, and **k**, statistical analysis was performed by two-
way ANOVA with correction for multiple comparisons. In **l**, statistical analysis was
performed by one-way ANOVA with Tukey's post hoc tests.

To determine whether bridge expression changed the phenotype
of CAR-T cells, we analyzed CAR-T cells on day 14 after CD3/CD28
bead-based T cell activation and classified them as naïve (CD45RA+
CCR7+), terminally differentiated effector ($T_{EMRA}$; CD45RA+ CCR7−),
central memory ($T_{CM}$; CD45RA− CCR7+), or effector memory T cells
($T_{EM}$; CD45RA− CCR7−) within the CD4 and CD8 CAR-T cell popula-
tions. There were no differences in cell phenotypes between the glyco-
and CD19-bridge CAR T cells (Supplementary Fig. 1h, i). Furthermore,
there was no significant difference in the overall cytokine secretion
between the two bridge conditions after 24-h stimulation with Capan-2
*C1GALT1* KO cells. However, TNF-α levels were modestly higher with
the TnMUC1 bridge, although this difference did not reach statistical
significance. (Supplementary Fig. 1j–m).

To test the hypothesis that the bridge molecule enhanced T cell
adhesion to the target cells, we measured the cell avidity of the Tn-
MUC1 bridge against the Capan-2 *C1GALT1* KO cell line. Cells were
seeded in z-Movi Cell Avidity Microfluidics chips to form a monolayer
on the surface. CAR-T cells with bridges were then introduced and

allowed to bind to the target cells. CAR-T cells were tracked, and the
binding avidity between target cells and CAR-T cells was quantified,
with gradient acoustic forces applied to disrupt the interaction
between the cells (Fig. 1j). We found that the Tn-MUC1 bridge sig-
nificantly increased cell avidity compared to the control CD19-bridge
against Capan-2 *C1GALT1* KO cells. This suggests that the enhanced
cytotoxic efficacy of the glyco-bridge CAR T cells in vitro is attributed
to the increased avidity between the CAR T cell and target cell
(Fig. 1k, l).

### Tn-MUC1 bridge CAR-T cells enhance tumor suppression in Capan-2 mouse models

To prepare to test this approach in vivo, we first examined the impact
of *C1GALT1* KO and itraconazole treatment on Capan-2 tumor growth.
Tumor cells ($5 \times 10^6$) were subcutaneously injected into NOD.Cg-
Prkdc^scid^ Il2rg^tm1Wjl^/SzJ (NSG) mice, and itraconazole was administered
via intraperitoneal injection at a dose of 40 mg/kg daily for 60 days.
This dosage was carefully determined to balance therapeutic efficacy

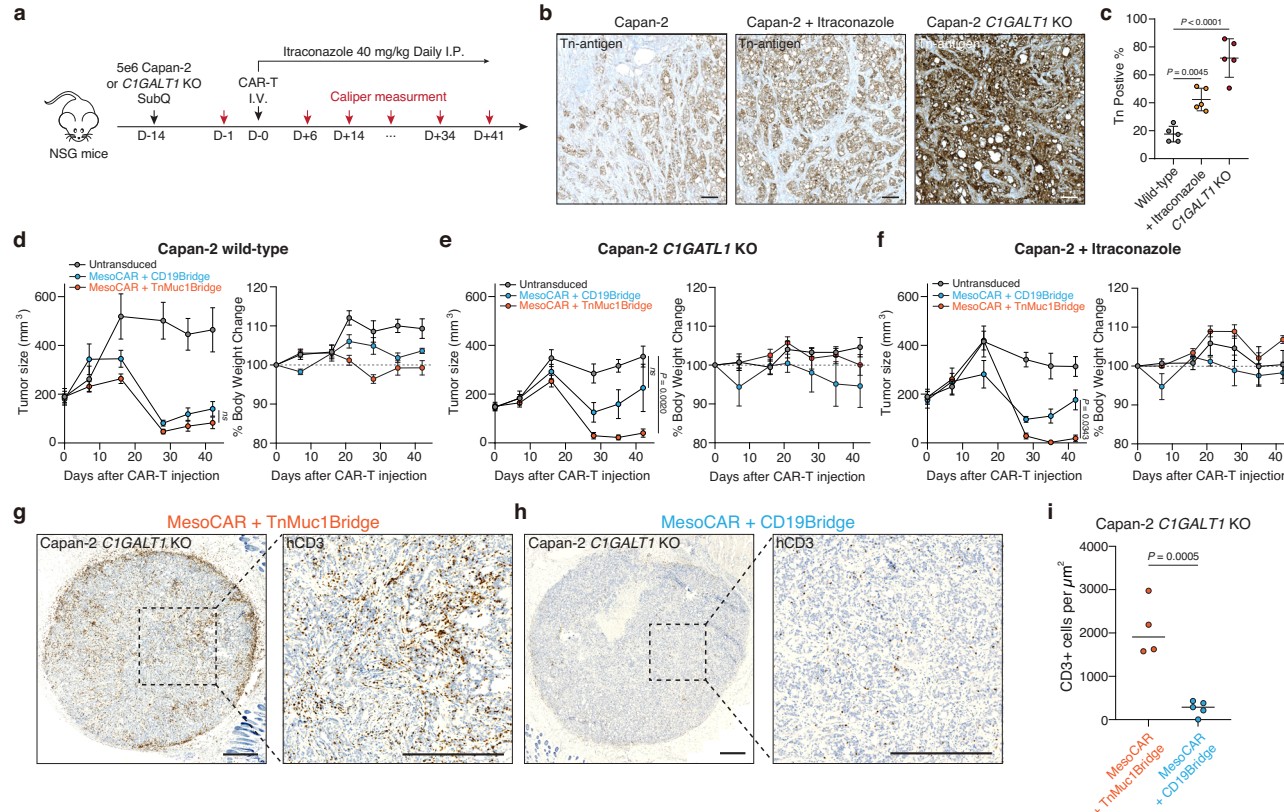

**Fig. 2 | In vivo validation of mesothelin CAR T cells with glyco-bridge. a** NOD.Cg-Prkdc[scid] Il2rg[tm1Wjl]/SzJ (NSG) mice were subcutaneously (SubQ) injected with $5 \times 10^6$ Capan-2 or Capan-2 *C1GALT1* KO cells on day −14. The mice were further treated intraperitoneally (I.P.) with itraconazole (40 mg/kg) once daily for 30 days. On Day 0, 5 mice per group received $2 \times 10^6$ CAR T cells for the Capan-2 *C1GALT1* KO group or $2.5 \times 10^6$ CAR T cells for the Capan-2 group. CAR-T cells are the indicated mesothelin-targeted CAR with glyco-bridge or untransduced T cell. **b** Representative IHC images for Tn antigen in formalin-fixed, paraffin-embedded specimens from mice injected with Capan-2, Capan-2 *C1GALT1* KO, and Capan-2 with itraconazole treatment. Scale bar, 10 µm. **c** Quantification of Tn antigen+ cells in tumors from each mouse. Results are mean ± s.d. of *n* = 5 mice. Tumor growth curve (left) and body weight change (right) for Capan-2 (**d**), Capan-2 *C1GALT1* KO

(**e**), and Capan-2 with itraconazole treatment (**f**). Results are mean ± s.e.m. of *n* = 5 mice. Representative IHC images for CD3 (T cell) in formalin-fixed, paraffin-embedded tumor specimens from mice injected with Capan-2 *C1GALT1* KO and mesothelin-targeted CAR-T cells with Tn-MUC1 bridge (**g**) or CD19-bridge (**h**). Scale bar, 500 µm. **i** Quantification of infiltrating CD3+ cells in tumors of mice treated with mesothelin-targeted CAR-T cell using either a Tn-MUC1 or CD19 bridging strategy. Data represent the mean for *n* = 4 for mesothelin-targeted CAR-T cells with Tn-MUC1 bridge and *n* = 5 for CD19-bridge, with data from tumor-cleared mice excluded. Statistical analysis was performed by two-tailed t-tests. In **c**, statistical analysis was performed by one-way ANOVA with Tukey's post hoc tests. In **d**–**f**, statistical analysis was performed by two-way ANOVA with correction for multiple comparisons.

and potential toxicity concerns during long-term treatment, corresponding to clinically relevant plasma concentrations (0.5–1.0 µg/mL) observed in human patients receiving standard oral itraconazole therapy[63–65]. Overall, no significant differences in tumor growth were observed among the conditions. (Fig. 2a and Supplementary Fig. 2a, b). Consistent with in vitro findings, in vivo itraconazole treatment increased Tn antigen expression in tumors, with a 2.4-fold increase compared to Capan-2 cells. Similarly, *C1GALT1* KO Capan-2 exhibited a 4.1-fold increase in Tn antigen expression compared to Capan-2 cells (Fig. 2b, c). We also confirmed that itraconazole had no effect on tumor growth at concentrations up to 10 µM in vitro, although Capan-2 *C1GALT1* KO cells exhibited slower in vitro growth rates compared to wild-type cells (Supplementary Fig. 2c).

Next, we measured CAR-T cell-mediated killing against Capan-2, Capan-2 *C1GALT1* KO, and itraconazole-treated Capan-2 in vivo. Capan-2 cell lines ($5 \times 10^6$) were injected subcutaneously into NSG mice. The tumors were monitored for 14 days until the average tumor size reached approximately 150–200 mm³. To determine the optimal dose, we tested mesothelin-targeted CAR-T cells at doses of $1 \times 10^6$ cells, $4 \times 10^6$ cells, and two separate infusions of $4 \times 10^6$ cells per week against Capan-2 *C1GALT1* KO cells. Based on these experiments, we identified the optimal dose range ($1–4 \times 10^6$ cells) for the Capan-2 model (Supplementary Fig. 2d). As expected, due to the relatively low levels of

MUC1 expression at baseline, no significant difference in tumor size was observed between the two bridges in the CAR-T cell condition in the Capan-2 wild-type model (Fig. 2d). However, in the Capan-2 *C1GALT1* KO model or the Capan-2 model treated with itraconazole, where MUC1 levels were increased, CAR-T cells with Tn-MUC1 bridge exhibited significantly more anti-tumor activity compared to the CD19-bridge (Fig. 2e, f and Supplementary Fig. 2e). Furthermore, no significant weight loss or poor body condition was observed (Fig. 2e, f). To further evaluate the effect of the glyco-bridge on CAR-T cell infiltration throughout the tumor, we stained human CD3+ cells in the residual tumors on Day 52 using an anti-human CD3 antibody. Notably, CAR-T cells with the Tn-MUC1 bridge showed approximately 8-times higher infiltration than the CD19-bridge into Capan-2 *C1GALT1* KO tumors (Fig. 2g–i). The tumor control by mesothelin-targeted CAR-T cells with the glyco-bridge strongly correlated with tumor infiltration by the T cells. Taken together, these results support the use of a glyco-bridge with CAR molecules to enhance anti-tumor activity.

## Activation of the CAR is dependent on recognition of the bridge's antigen

Since we observed no significant difference in efficacy between the glyco- and CD19-bridge in tumors with relatively low Tn-MUC1 expression or CD19 expression, we next investigated how the

expression level of the bridge target antigen contributes to CAR-T cell-mediated killing. As a working hypothesis, we tested four conceptual models of CAR-T cell-mediated killing based on the density of the antigen recognized by the bridge molecule, rather than the CAR itself: independent, insensitive, linear, and ultrasensitive. These models reflect different hypothetical dependencies of CAR-T cell activity on bridge antigen density. (Fig. 3a). As a model, we utilized the RPMI 8226 multiple myeloma cell line, which lacks expression of mesothelin and CD19 (Supplementary Fig. 3a, b) and has moderate expression levels of MUC1 (Fig. 3b). This made it an ideal system to test the contribution of the glyco-bridge, since CAR-T cells targeting mesothelin would not engage the tumor cells directly. Similar to Capan-2 cells, RPMI 8226 cells exhibited increased surface expression of Tn-MUC1 and Tn antigens upon itraconazole treatment, enabling us to assess how modulation of bridge antigen density impacts CAR-T cell activity. (Fig. 3b and Supplementary Fig. 3c). Using Tn-MUC1 bridge mesothelin-targeted CAR T cells against the RPMI 8226 cell line, there was no significant cytotoxicity, even with itraconazole treatment, suggesting that the Tn-MUC1 bridge was not involved in CAR-mediated killing when the expression of the CAR antigen was low (Fig. 3c). This suggested that sufficient activation of the CAR is necessary for the bridge to contribute to effective cytotoxicity. To verify this result with our other bridge molecule targeting CD19, we again used the RPMI8226 cells but overexpressed CD19 and compared them to wild-type cells with low CD19 expression. This time, the mesothelin-targeted CAR-T cells with the CD19-bridge exhibited increased cytotoxicity when CD19 was overexpressed (Fig. 3d–g). The same trend was observed in Jeko-1 cells, which similarly express low mesothelin but high levels of CD19 in WT cells (Supplementary Fig. 3a, b). Conversely, we knocked out CD19 from Jeko-1 cells by CRISPR/Cas9, then re-expressed CD19 by transduction generated clones with varying CD19 expression levels by single-cell sorting. The number of CD19 molecules per cell was quantified using the BD Quantibrite™ Kit (Fig. 3h). In Jeko-1 cells, CD19-bridge-mediated cytotoxicity was abrogated when CD19 was knocked out, demonstrating that the cytotoxicity was dependent on expression of the antigen targeted by the bridge molecule (Fig. 3i, j). Our data revealed a clear power-law dependence of CD19-bridge-mediated cytotoxicity on CD19 density in Jeko-1 cells, with a scaling exponent of −0.74 (Fig. 3k). Similarly, CD19 overexpression in RPMI-8226 multiple myeloma cell lines exhibited comparable trends, with a scaling exponent of −0.16 (Fig. 3g). Thus, these findings suggest that additional binding via the bridge molecule may contribute to CAR-T cell cytotoxicity even in the presence of minimal CAR antigen expression, raising the possibility of partial CAR target-independent killing.

To further explore this CAR target-independent killing, we altered the different functional domains of the CAR and the bridge to determine which domains were necessary for T cell killing. We generated an scFv-deleted CAR, a complete CAR deletion, and a CD3ζ deleted-CAR. We also substituted the CD28 H/TM domain with the PDGFRβ TM domain to investigate the linkers interaction and the interaction between the endogenous TCR and CD28 or CD8. Additionally, we replaced the inactive CD28 domain with the NGFR ICD domain, which is unrelated to CAR signaling (Fig. 3l and Supplementary Fig. 3d). We used each of these CAR T cells against WT and CD19-overexpressing RPMI 8226 cells. Similar to what we had observed previously, there was no difference in the killing of WT RPMI 8226 cells. With CD19-overexpressing RPMI 8226 targets, only CAR-T cells containing intracellular signaling domains showed enhanced killing, confirming that bridge-mediated killing is CAR-dependent. Importantly, CAR constructs lacking the scFv still showed cytotoxicity in the presence of the CD19 bridge, suggesting that bridge can activate CAR signaling independently of direct antigen recognition by the CAR scFv (Fig. 3m and Supplementary Fig. 3e–h). Fluorescent microscopy also revealed no direct killing by the bridge alone, but it did show aggregation of Jeko-1 cells in the presence of the CD19-bridge. This clustering is consistent

with bridge-mediated cell adhesion, likely resulting from bridge molecules binding to CD19 on neighboring target cells (Supplementary Fig. 3i). Based on these results, we identified two main interactions between the CAR molecule and the bridge molecule (Fig. 3n). Previous studies have shown that scFv aggregation occurs through VH/VL interactions alone, without protein unfolding, and this phenomenon is currently considered in CAR manufacturing to minimize scFv interactions[66,67]. Likewise, the bridge molecule can activate the CAR molecule to kill target cells.

We showed that CAR-mediated killing does not occur via the Tn-MUC1 bridge in Fig. 3c, whereas CAR-mediated killing can occur in a target-independent manner through the CD19-bridge. This finding led us to examine the affinity of the two scFvs. The dissociation constant of the FMC63 scFv (αCD19) ranges from 300 pM to 5 nM[68–70], whereas the 5E5 scFv (αTn-MUC1) ranges from 2 to 4 nM[71,72]. This difference suggests that the ligand affinity, in addition to the antigen density, could play a role in the efficacy of killing mediated by the bridge. We hypothesize that despite their ability to engage target glycans, low-affinity binders like lectins may not provide sufficient binding strength to trigger CAR-mediated killing through the bridging interaction (Fig. 3o). Therefore, this approach would avoid undesirable CAR T cell activation in response to the bridge target alone, since lectins are also expressed in normal tissue (albeit at lower levels).

## HPA lectin-based CAR-T cells expand the targeting scope of Tn antigens

To universally target Tn antigen structures beyond Tn-MUC1, we leveraged the HPA lectin as the antigen-binding domain of the CAR molecule. Unlike MUC1-specific binders, HPA recognizes the Tn antigen motif (GalNAcα1-Ser/Thr) across a wide range of glycoproteins, including mucins such as MUC5AC and MUC16 in pancreatic cancer[73]. This enables broader detection of Tn antigen-expressing tumor cells regardless of the specific mucin backbone. This lectin has a relatively low dissociation constant while maintaining high specificity for Tn antigen. The dissociation constant of HPA with GalNAc is approximately 130 μM[74], which is lower than that of conventional FMC63 scFv. We firstly aimed to quantitatively evaluate the effectiveness of HPA as a binder for CARs. As previously reported[44,75], the Tn structure is predominantly expressed in the ductal regions of pancreatic ductal adenocarcinoma[76,77]. We confirmed that HPA binds to these regions using HPA-biotin on a human pancreatic tissue microarray (PA807). H-score analysis showed significantly elevated HPA-biotin binding in PDAC tissues relative to normal pancreas, indicating increased expression of Tn antigen in PDAC patients. (Fig. 4a–c). Furthermore, when we inhibited C1GALT1 enzymatic function using itraconazole, HPA binding to Capan-2 cells increased compared to baseline (Fig. 4d). We then engineered a HPA-based CAR by replacing the scFv binder with HPA to determine its efficacy as a binding component (Fig. 4e). Since wild-type HPA has a hexagonal structure formed by two trimeric subunits, we designed a dual HPA (HPAx2) and triple (HPAx3) structure to enhance binding affinity. Each HPA unit is fused via a $(G_4S)_3$ linker (Fig. 4e).

To quantify the binding affinity of each HPA-based CAR, we measured cell avidity using the z-Movi Cell Avidity Microfluidics System against Capan-2 wild-type cells (low Tn antigen) and Capan-2 C1GALT1 knockout cells (high Tn antigen). We observed that CARs with more HPA units exhibited higher avidity toward cells with elevated Tn antigen expression (Fig. 4f, g). However, lentiviral transduction efficiency was markedly reduced for HPAx3 CAR constructs due to the increased transgene size (Supplementary Fig. 4a). We further measured CD69 expression on CAR-T cells after a 24-h co-culture with Capan-2 and Capan-2 C1GALT1 KO cells. The dual HPA structure led to higher activation and improved in vitro killing specificity against Capan-2 C1GALT1 KO cells compared to the single HPA structure (Supplementary Fig. 4b).

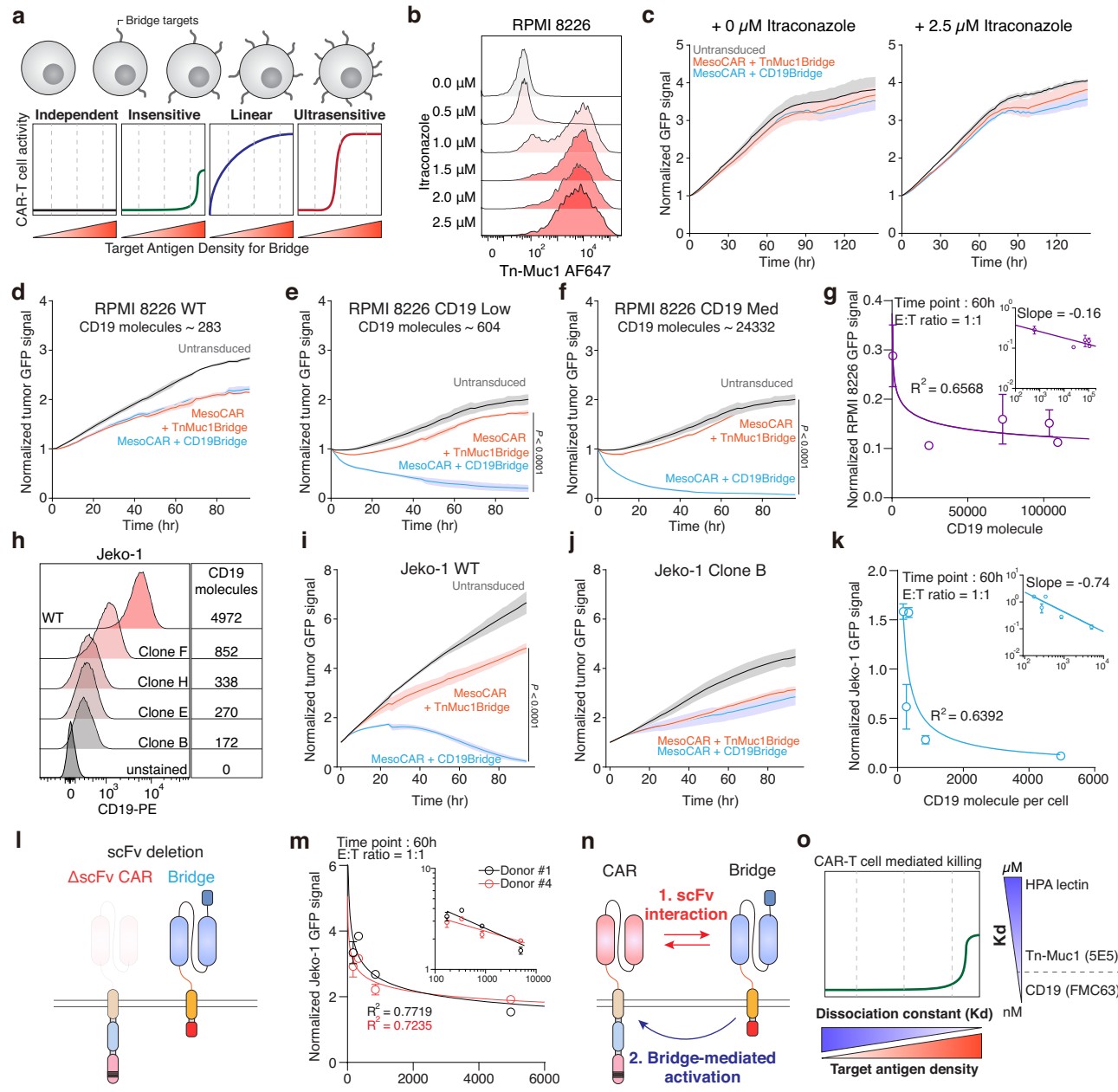

**Fig. 3 | Glyco-bridge activates CARs in a target antigen density-dependent manner. a** Schematic showing the mechanism of glyco-bridge depending on the target antigen density of the glyco-bridge. **b** Surface level of Tn-MUC1 on RPMI 8226 cells following itraconazole incubation for 48 h. **c** Real-time cytotoxicity assay against RPMI 8226 cells with itraconazole with indicated bridges at a 1:1 E:T ratio. Results are mean ± s.d. of $n = 3$ independent measurements. **d–f**, Real-time cytotoxicity assay with mesothelin-targeted CAR-T cells with indicated bridges at a 1:1 E:T ratio. Panels show RPMI 8226 (**d**), low CD19 (**e**), and medium CD19 (**f**). Results are mean ± s.d. of $n = 3$ independent measurements. **g** log of the number of CD19 molecules versus log of the normalized target cell GFP signal from **d–f** at time point 60 h; best-fit line to a power-law model with slope = −0.16 and $R^2 = 0.6568$. **h** Surface level of CD19 on Jeko-1 clones. **i**, **j** Real-time cytotoxicity assay with mesothelin-targeted CAR-T cells with indicated bridges at a 1:1 E:T ratio. Panels show Jeko-1 (**i**) and Jeko-1 clone B (**j**). Results are mean ± s.d. of $n = 3$ independent measurements. **k** log of the number of CD19 molecules versus log of the normalized target cell GFP signal from (**i, j**) at time point 60 h; best-fit line to a power-law model with slope = −0.74 and $R^2 = 0.6392$. **l** Schematic showing ΔSS1 scFv on CAR molecules. **m** log of the number of CD19 molecules versus log of the normalized GFP signals from Jeko-1 cell lines from (**h**) using ΔSS1 scFv CAR at time point 60 h; best-fit line to a power-law model for each donor; $R^2$ for 0.7719 (donor #1) and 0.7235 (donor #4). Results are mean ± s.d. of $n = 3$ independent experiments per donor. **n** Proposed relationship between CAR and bridge molecules. **o** Proposed mechanism between glyco-bridge-mediated killing, target antigen density for the glyco-bridge, and binding affinity. In **b**, **h** representative flow cytometry data from three independent experiments. In **e**, **f**, and **i** statistical analysis was performed by two-way ANOVA with correction for multiple comparisons.

We next utilized a non-immortalized pancreatic ductal adenocarcinoma patient-derived xenograft (PDX) cell line (PDX1294), which has high expression levels of both Tn antigen and mesothelin (Fig. 4h–j and Supplementary Fig. 4e). The expression level of Tn antigen on PDX1294 was higher than Capan-2 *C1GALT1* KO cells or itraconazole-treated Capan-2. Confocal images confirmed that PDX1294 bind more HPA on its cell membrane than Capan-2 cells (Fig. 4h–j). The dual HPA-directed CAR-T cells exhibited enhanced cytotoxicity against PDX1294 cells than single HPA in vitro (Fig. 4j and Supplementary Fig. 4c, d). We then compared HPA-directed CAR-T cells with Tn-MUC1 CAR-T cells

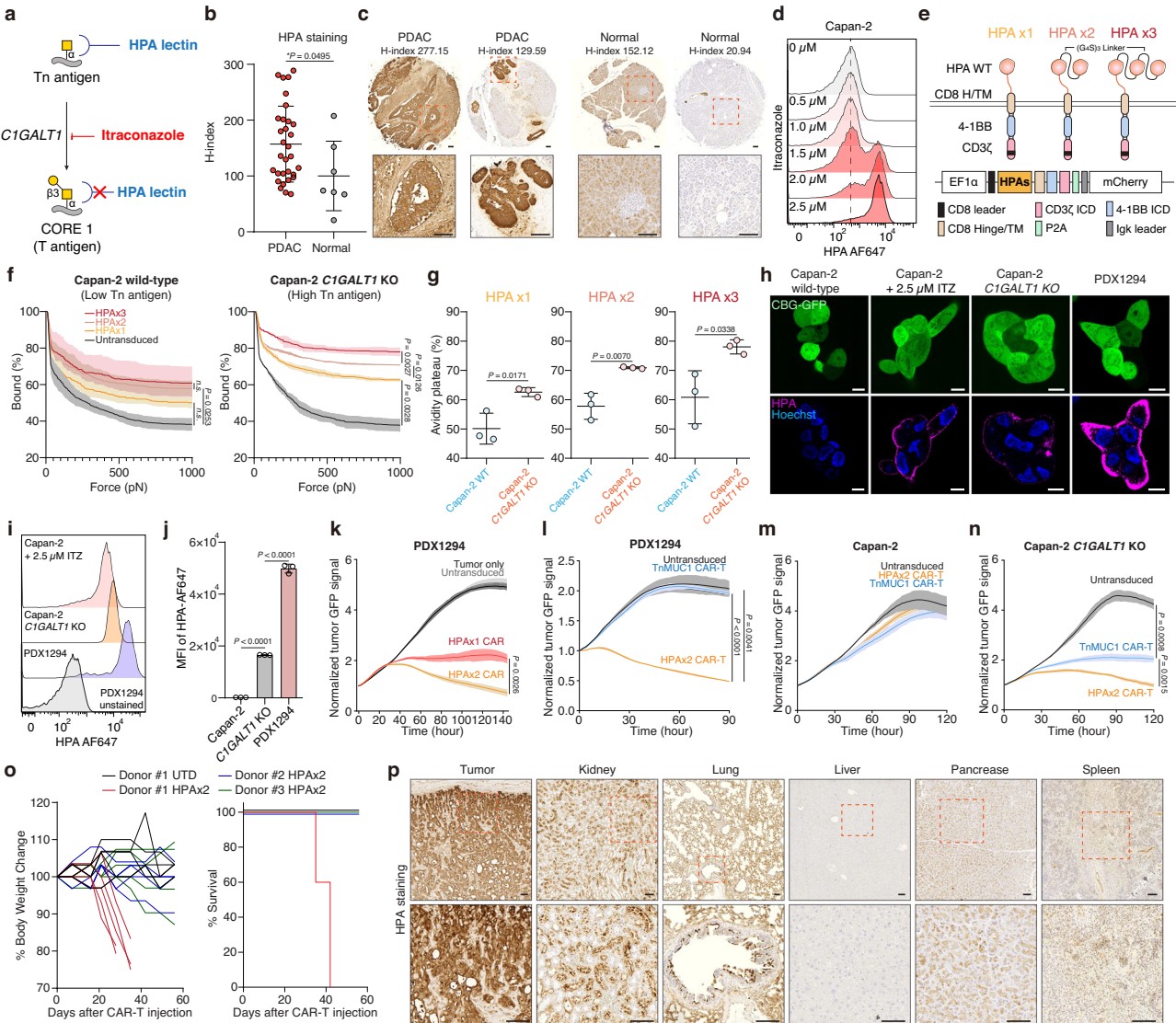

**Fig. 4 | Snail lectin HPA-based CAR T cells show antitumor activity and associated toxicity against Tn antigen-positive pancreatic tumor cell line.**
**a** Biosynthetic pathway for Tn antigen. **b** HPA staining quantified by *H*-score in PDAC versus normal pancreatic tissues (PA807; $n = 31$ and 7). *H*-score represents the weighted sum of cells with low (0.14–0.4), moderate (0.4–0.6), or strong (>0.6 OD) intensity. Significance was assessed by unpaired two-tailed t-test.
**c** Representative IHC images of HPA-stained tissue sections from b. Scale bar, 100 μm. **d** Surface level of HPA binding on Capan-2 following itraconazole incubation at the indicated concentrations for 48 h. **e** Schematic showing HPA-based CAR T cells. **f** Strength of interaction between target cells and indicated HPA CAR-T cells. Percentage of total CAR-T cells remaining bound to target cells as the acoustic force ramp is applied from 0 to 1000 pN are shown. Results are mean ± s.e.m. of $n = 3$ independent measurements. **g** Percentage of CAR-T cells remaining bound to the target cells at the avidity plateau under 1000 pN force from Fig. 4f. Significance

was assessed by unpaired two-tailed t-test. **h** Representative confocal images of HPA-stained cells. Scale bar, 10 μm. **i** Surface level of HPA binding on indicated target cells. **j** HPA binding on each cell from (**g**). Results are mean ± s.d. of $n = 3$ independent measurements. Real-time cytotoxicity assay against PDX1294 (**k**, **l**), Capan-2 (**m**), Capan2- *C1GALT1* KO (**n**) target cells with indicated CAR T cells at 1:1-2:1 E:T ratio. Results are mean ± s.d. of $n = 3$ independent measurements. **o** NSG mice implanted s.c. with $5 \times 10^6$ Capan-2 *C1GALT1* KO cells received $2 \times 10^6$ dual-HPA CAR-T or UTD cells on day 0 (5 mice/group; $n = 3$ donors). Body weight change (left) and survival percentage for $n = 5$ mice. **p** Representative HPA staining of tumor and major organs from o ($n = 5$ mice). Scale bar, 100 μm. In (**d**, **i**) representative flow cytometry data from three independent experiments. In **f** and **k**–**n** statistical analysis was performed by two-way ANOVA with correction for multiple comparisons. In (**j**), one-way ANOVA with Tukey's post hoc tests.

using PDX1294 and Capan-2 cell lines. HPA-directed CAR-T cells exhibited superior killing compared to Tn-MUC1 CAR-T cells, even in Capan-2 *C1GALT1* KO cell lines, which have higher Tn-MUC1 expression (Fig. 4k–n). Using PDX1294 cells, we further evaluated triple HPA constructs to determine whether they outperform the dual HPA design. When normalized for CAR expression, the killing efficiency of the triple construct was comparable to that of the dual HPA CAR.

To explore alternative lectins that recognize Tn or sialyl-Tn (STn) antigen antigens, we next compared the performance of HPAx2 with other lectins, including the human macrophage galactose N-acetyl-

galactosamine-specific lectin (MGL) and sialic acid-binding immunoglobulin-like lectin 15 (Siglec-15). MGL binds to the Tn antigen, while Siglec-15 preferentially recognizes the tumor-associated STn antigen[78,79] (Supplementary Fig. 5a). Using fluorescently labeled recombinant MGL and Siglec-15, we confirmed strong binding to PDX1294 cells, indicating overexpression of both Tn and STn antigens in this model (Supplementary Fig. 5b). We then tested CAR-T cells engineered with these lectins in cytotoxicity assays. While Siglec-15 CAR-T cells exhibited minimal killing activity, MGL-based CAR-T cells showed effective cytotoxicity, with the MGLx1 construct superior

killing than the dual MGL (MGLx2). However, MGLx1 CARs were still less effective than the dual HPAx2 CARs (Supplementary Fig. 5c, d). Given these comparative results, along with the large transgene size and reduced viral transduction efficiency of HPAx3, we selected the dual HPA CAR as the optimal configuration.

We then tested these dual-HPA-directed CAR-T cells in vivo using Capan2 *C1GALT1* KO model. In NSG mice, HPA-directed CAR-T cells significantly reduced tumor size in Capan-2 *C1GALT1* KO tumors (Supplementary Fig. 5g, h). However, in one group of mice treated with CAR T cells from Donor #1, we observed toxicity that resulted in rapid weight loss and decreased survival of these mice (Fig. 4o). We observed a similar toxicity in 1/5 mice from each of the other two donor groups, suggesting an on-target, off-tumor toxicity from the dual-HPA CAR T cells. To further investigate this toxicity, we harvested tumors (Capan-2 *C1GALT1* KO) and major organs, including the kidney, lung, liver, pancreas, and spleen, from the mice and performed HPA staining. We observed notable HPA binding in the kidney and lung, suggesting potential on-target, off-tumor recognition in these tissues (Fig. 4p). This observation prompted us to repurpose the dual-HPA module as a glyco-bridge rather than using it as the primary antigen-binding domain of the CAR. Given its effective recognition of Tn antigens, HPA can serve as a modular attachment to facilitate tumor targeting without directly mediating CAR activation. These findings support the use of HPA lectins as auxiliary binders that enhance CAR function while minimizing on-target, off-tumor toxicity.

### Enhancing CAR-T cell efficacy with dual HPA bridge in PDX models

We next explored the hypothesis that using a dual HPA structure as a glyco-bridge could broaden Tn antigen targeting and enhance CAR-T cell efficacy. We modified the existing glyco-bridge with the dual HPA structure (Fig. 5a–c). The expression level of CAR with HPA glyco-bridge was similar to the levels of the Tn-MUC1 bridge and CD19-bridge, as measured by $(G_4S)_3$ MFI (Supplementary Fig. 6a). To confirm that the dual HPA glyco-bridge does not induce killing, we used Jurkat cells, which overexpress Tn antigen but lack mesothelin expression (Supplementary Fig. 6b), as a control model. Even with high levels of HPA binding, glyco-bridge-mediated cytotoxicity did not occur (Supplementary Fig. 6c, d).

To better evaluate this bridge-based approach, we next utilized a panel of non-immortalized patient-derived xenograft (PDX) cell lines, including the previously characterized PDX1294 and two additional models, PDX1275 and PDX1319. We first compared the expression levels of mesothelin, HPA-binding glycans, MUC1, and Tn-MUC1 across these models (Fig. 5d). PDX1294 exhibited high levels of both mesothelin and HPA-binding glycans. PDX1275 showed moderate mesothelin expression and high MUC1 expression, closely resembling the Capan-2 cell line. In contrast, PDX1319 expressed low levels of mesothelin, MUC1, and HPA-binding glycans (Fig. 5d). Using this diverse panel of PDX models, we compared the cytotoxic function of CAR-T cells equipped with either HPA-bridge or Tn-MUC1-bridge modules. As expected, PDX1294 responded more strongly to the HPA-bridge CAR-T cells, while PDX1275 was more effectively targeted by Tn-MUC1-bridge CAR-T cells. In PDX1319, limited antigen expression correlated with minimal response to both bridge strategies, including the conventional mesothelin-targeting CAR (Fig. 5e). While our shift to HPA bridge was intended to broaden the glycan recognition landscape beyond MUC1, we found that Tn-MUC1-specific constructs retained strong activity, particularly in models with high Tn-MUC1 expression but low HPA reactivity, such as PDX1275. Based on these observations, we examined whether the HPA bridge enhances CAR-T cell avidity in PDX1294, which overexpresses the Tn antigen. Similar to the Tn-MUC1 bridge shown in Fig. 1l–n, mesothelin-targeted CAR-T cells with the HPA bridge exhibited higher avidity compared to the CD19 bridge control (Fig. 5f).

We next tested whether a dual HPA-based bridge could reduce the toxicity of HPA-CAR. Using the same donors, we conducted experiments in the Capan-2 *C1GALT1* KO subcutaneous model, which was previously used for the HPA-CAR experiment in Fig. 4o, p (Fig. 5g). Interestingly, the toxicity associated with HPA-CAR was no longer present when the HPA-binding moiety was used as a bridge instead of as a CAR. Additionally, the mesothelin-targeted CAR with the HPA-bridge resulted in improved overall survival and induced either similar or greater tumor control compared to the HPA-CAR, depending on the donor (Fig. 5h–m). To explore the difference in toxicity and try to understand why the HPA-CAR was toxic, we harvested tissues from mice treated with the HPA-CAR versus the mesothelin-targeted CAR with the HPA-bridge. We found that HPA-CARs exhibited nonspecific infiltration into the kidney, liver, and lung, whereas the HPA-based bridge did not infiltrate into these organs and was primarily found in the tumor tissue (Fig. 5n and Supplementary Fig. 7). To validate this approach in another tumor model, we used a pancreatic ductal adenocarcinoma patient-derived xenograft. PDX1294 cells ($5 \times 10^6$) were injected subcutaneously into NSG mice, and tumors were monitored for 14 days; $2 \times 10^6$ CAR-T cells were injected intravenously when the average tumor size reached approximately 200 mm³. After a single injection of CAR-T cells, tumor size was monitored via caliper measurement (Fig. 5o). We confirmed that CAR-T cells with a dual HPA bridge exhibited superior tumor-killing and extended survival compared to CAR-T cells with the control CD19-bridge (Fig. 5p–r and Supplementary Fig. 6e). These results demonstrate that the HPA bridge enhances CAR-T cell efficacy by improving tumor recognition and cytotoxicity in Tn antigen-rich PDACs.

Taken together, these data suggest that adding the glyco-bridge to CAR-T cells can overcome the glycocalyx barrier and increase target binding and killing. Moreover, the bridge system's ability to activate CAR-T cells based on antigen density and binding affinity was validated. By using HPA lectin, which specifically recognizes Tn antigens regardless of their carrier glycoproteins, we incorporate a HPA lectin as a binder to the glyco-bridge module that not only improves killing efficacy, even in a pancreatic cancer PDX model, but also significantly reduces the toxicity observed from using the same binder in CAR format.

## Discussion

While previous studies have explored dual CAR or a costimulatory receptor (CCR) system to enhance cytotoxicity via increased avidity[57,58,80], our study demonstrates that equipping CAR-T cells with a Tn-MUC1 binder and Tn antigen binder enables direct targeting of the cancer cell glycocalyx, significantly enhancing their efficacy in pancreatic cancer models in vitro and in vivo. Unlike prior strategies that focus on protein antigens, our glyco-bridge approach uses a non-signaling glycan-binding module to interact with tumor-associated glycans, thereby enhancing the functional avidity of CAR-T cells through additional glycan or mucin engagement. To better understand the underlying mechanisms of enhanced CAR-T cell activity, we investigated how the glyco-bridge enhanced CAR-T cell efficacy. Our data suggest that the glyco-bridge increased cell avidity and improved tumor penetration, offering an approach to engage the cancer cell glycocalyx without directly initiating cytotoxicity. To broaden the targeting capabilities against a wider range of cancer-associated glycans, such as the Tn antigen, we developed a binder derived from an existing lectin called HPA. The tandem HPA lectins were applied in a glyco-bridge system, demonstrating strong specificity in Tn antigen-overexpressing tumor models, further enhancing the efficacy of CAR-T cells. While prior studies have shown that dual CAR or CAR/CCR systems can enhance cytotoxicity via increased avidity, our work takes a distinct approach by using a glycan-binding module to engage the tumor glycocalyx. This approach is designed to investigate how glycocalyx recognition contributes to immune synapse formation and T

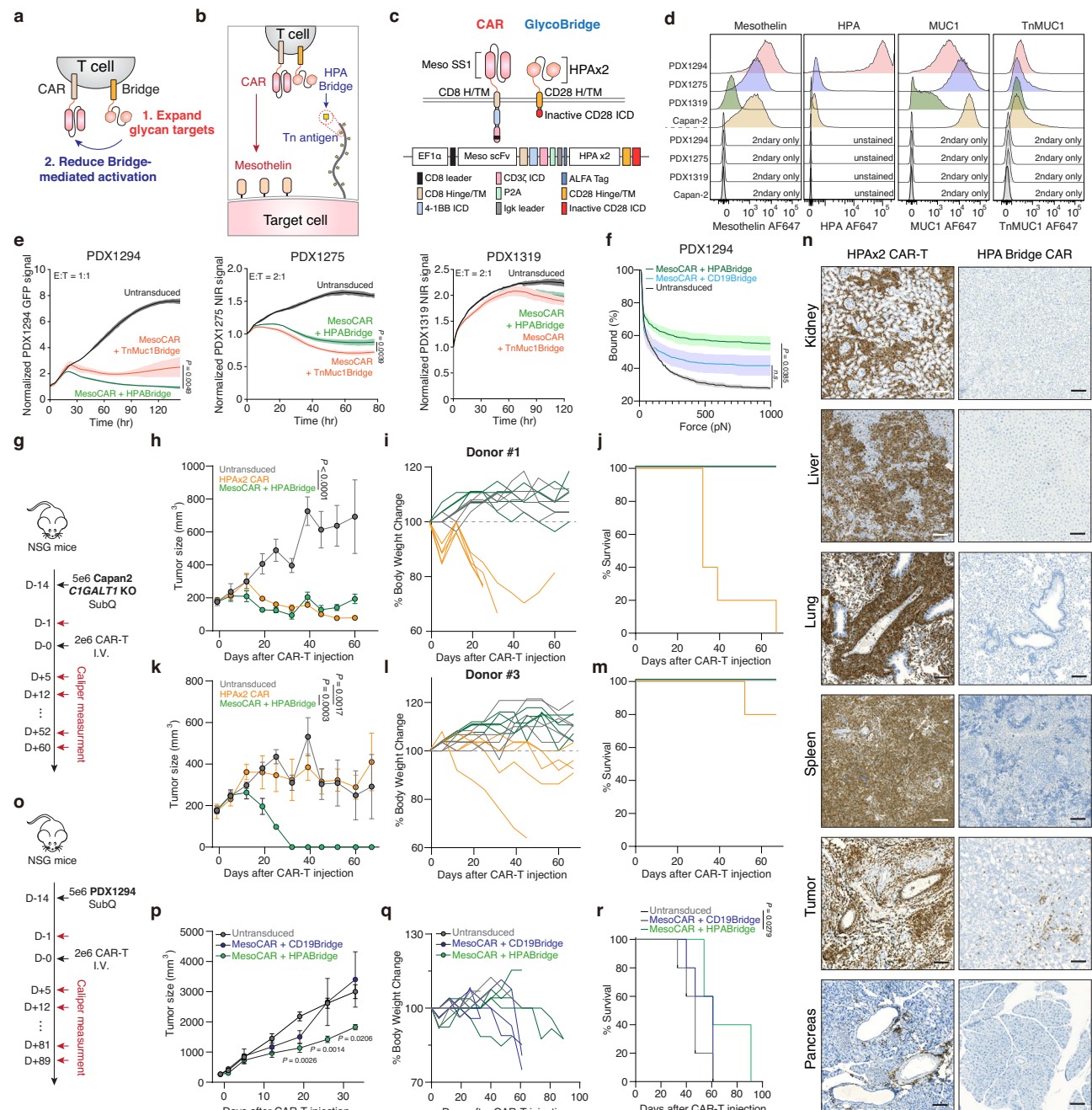

**Fig. 5 | Glyco-bridge with tandem HPAs show antitumor activity against pancreatic cancer patient-derived xenograft model. a** Schematic representation of CAR-T cells with a tandem HPA bridge, showing expanded glycan targeting and reduced bridge-mediated activation. **b** Schematic of mesothelin-targeting CAR T cells with tandem HPAs-based glyco-bridge. **c** Graphic representation of CAR and bridge constructs. **d** Flow cytometry analysis of mesothelin, HPA, MUC1, and Tn-MUC1 expression on Capan-2 and PDX-derived cells from of $n = 3$ independent experimental replicates. **e** Real-time cytotoxicity assay against indicated PDX-derived cells with indicated CAR T cells at 1:1-2:1 E:T ratio. Results are mean ± s.d. of $n = 3$ independent measurements. **f** Strength of interaction between PDX1294 and indicated CAR-T cells. Percentage of total CAR-T cells remaining bound to target cells as the acoustic force ramp is applied from 0 to 1000 pN are shown. Results are mean ± s.e.m. of $n = 3$ independent measurements. **g** NSG mice were subcutaneously injected with $5 \times 10^6$

Capan2 *C1GALT1* KO cells on day −14. On Day 0, 5 mice per group received $2 \times 10^6$ CAR-T cells with HPA bridge or HPA-directed CAR-T cell or Untransduced T cell (UTD). Tumor growth curve (**h**), body weight change (**i**) and survival percentage (**j**) for Capan2 *C1GALT1* KO model using Donor #1. Results are mean ± s.e.m. of $n = 5$ mice. Tumor growth curve (**k**), body weight change (**l**) and survival percentage (**m**) for Capan2 *C1GALT1* KO model using Donor #2. Results are mean ± s.e.m. of $n = 5$ mice. **n** Representative IHC images for human CD3 staining across different organs from Fig. 5g–m ($n = 5$ mice). Scale bar, 100 μm. **o–r**, NSG mice were subcutaneously injected with $5 \times 10^6$ PDX1294 cells on day −14. On Day 0, 5 mice per group received $2 \times 10^6$ CAR-T cells with HPA bridge or CD19-bridge or (UTD). Tumor growth curve (**p**), body weight change (**q**) and survival percentage (**r**) for PDX1294 cell model. Results are mean ± s.e.m. of $n = 5$ mice. In **e**, **f**, **h**, **k**, **p**, and **r** statistical analysis was performed by two-way ANOVA with correction for multiple comparisons.

cell function, particularly in mucin-rich tumors. Unlike previous designs, the glyco-bridge targets tumor-associated glycan structures without intracellular signaling, allowing us to isolate the effects of glycan engagement on CAR-T activity. While our current study focused

on a non-signaling glycan-binding bridge to isolate the role of glycan-mediated adhesion, future work could explore incorporating intracellular costimulatory domains (e.g., CD28 or 4-1BB) into the glyco-bridge module. These modules may further potentiate CAR-T cell

activation, as suggested by previous studies using chimeric costimulatory receptors.

These glyco-bridges offer insight into the dual antigen-targeting mechanisms. Specifically, our results suggest that the interaction between two CAR molecules extends beyond a simple OR-gate function. Rather, interactions among CAR molecules can influence the activation dynamics of dual or multiple CARs. Based on our observations using a CAR and bridge, this relationship may depend on antigen density and binding affinity. Furthermore, these findings suggest that even binding molecules lacking intracellular signaling domains (ICDs) can enhance CAR-T cell activity in the presence of a complete CAR. This concept is reminiscent of prior work by Haubner et al.[57] and Katsarou et al.[58], where signaling-incompetent CARs (lacking CD3ζ) could contribute to activation when co-expressed with a second, signaling-competent CAR through shared transmembrane domain-mediated dimerization. However, our study shows that bridge-mediated binding can still activate CAR ICDs in the absence of other transmembrane domains, suggesting an alternative mechanism of receptor clustering or synapse stabilization that does not rely on CAR-CAR dimerization.

While we did not observe glyco-bridge-mediated activation of CAR-T cells, likely due to the relatively low binding affinity of the HPA lectin, our findings with the CD19-bridge highlight the importance of considering how the characteristics of the bridge molecule itself, such as binding affinity, dissociation rate, and binding force, influence on CAR activation. These principles are well-established in the context of scFv-based CAR design, where high-affinity interactions can enhance target binding but may also lead to off-tumor toxicity or T cell exhaustion. Our work suggests that similar considerations apply to bridge-mediated CAR activation strategies, wherein the choice of binder must balance engagement strength with specificity and safety. This balance is particularly relevant in modular CAR systems, such as IF-better gates[57], where precise control over CAR activation critically depends on the binding properties of the additional binding molecule. Understanding how binding parameters influence CAR activation will be important for further optimizing modular CAR-T systems. Future studies incorporating bridge molecules with defined affinity differences, such as CD19 binders with varying kinetic properties, may help understand how these features modulate activation thresholds.

Importantly, our study did not directly target the Tn antigen with a CAR but rather leveraged it indirectly to enhance immune cell accessibility. By mimicking the Tn antigen structure through the knockout of T-synthase or itraconazole treatment, we observed elevated Tn antigen expression in cell lines that was comparable to actual patient cancer cells, as demonstrated in PDX1294. This approach underscores the potential for modulating glycosylation pathways to improve immune recognition of cancer cells. However, the precise molecular mechanisms driving increased expression of COSMC and T-synthase in Tn-positive tumors remain poorly understood. Elucidating these mechanisms could reveal novel molecular targets to combine with CAR-T cells, paving the way for more refined and effective cellular therapies. Furthermore, exploring the regulation of glycosyltransferase activity in Tn-positive tumor environments would help identify additional biomarkers or therapeutic strategies to target cancer cells, particularly for cancers with high glycan heterogeneity.

In the context of CAR-T strategies that target glycans via lectins, previous approaches have primarily focused on specific glycans, such as mannose (via the H84T banana lectin[81]) and glycoproteins like GD2[82] and GD3[83], some of which are currently in clinical development. While lectin-based CAR-T cell approaches have shown effectiveness in targeting the limited and complex glycan structures, challenges remain regarding binding affinity, selectivity, and toxicity. Introducing multimeric structures of lectins or finding essential mutations that can increase the binding specificity of lectins could enhance both the specificity and affinity of CAR-T cells for glycan targets. Furthermore, combining lectin-based CAR-T strategies with glycoengineering techniques, such as the use of glycosylation inhibitors or glycosylation-related gene editing to manipulate the glycosylation patterns of target cells, could help address the glycan heterogeneity in tumor microenvironments. In addition to these approaches, enzymatic degradation of the glycocalyx using mucin-degrading enzymes such as StcE offers a complementary strategy to enhance CAR-T cell accessibility. Our previous work demonstrated that mucinases can improve immune cell recognition and CAR-T cytotoxicity in mucin-rich cancer cells[17,19,20,84]. However, potential safety concerns regarding mucinase toxicity and off-target effects should be further investigated and carefully evaluated in future applications.

Moreover, HPA lectin binds to a wide range of glycosylated proteins, beyond just MUC1, making it a valuable tool for targeting cancers with high Tn antigen expression, including pancreatic, lung[41], stomach[40], bladder[35], ovarian[36], cervical[85,86], colon[76,87], and prostate tumors[38]. This broad applicability, when combined with the glyco-bridge system, has the potential to improve the effectiveness of current CAR-T cell therapies for a variety of cancer types. Importantly, the broad binding profile of HPA also raises the possibility of off-target recognition, particularly in normal tissues with low levels of Tn antigen. Careful assessment of binding specificity and context-dependent reactivity, ideally across a broad panel of human tissues and in immunocompetent models, will be important in future applications. Moreover, given that mesothelin is one of the key CAR targets in pancreatic models, its translational relevance depends on whether antigen distribution in mice reflects that in humans. Prior studies have shown that the expression pattern, biochemical characteristics, and tissue distribution of mesothelin in mice are similar to those in humans[88,89]. Combining mesothelin-targeted CARs with the glyco-bridge system may enhance the therapeutic potential of this approach. Furthermore, while our study specifically addresses glycocalyx-mediated barriers to CAR-T cell engagement, we recognize that pancreatic ductal adenocarcinoma presents additional immunosuppressive challenges, including a dense stromal matrix, immune checkpoint signaling, and metabolic constraints. Future work may be needed to integrate glyco-targeting strategies with other modalities that counteract these immunosuppressive mechanisms to achieve more durable therapeutic responses in PDAC.

## Methods

### Primary cells, Cell lines, and PDX tumor

Capan-2 (ATCC; HTB-80), RPMI 8226 (ATCC; CCL-155), and Jurkat (ATCC; TIB-152) cells were cultured in RPMI 1640 media (Thermo Fisher Scientific; Cat# 72400047) supplemented with 10% fetal bovine serum (Thermo Fisher Scientific) and 1x penicillin/streptomycin (Thermo Fisher Scientific) at 37 °C in 5% $CO_2$. Human T cells were isolated (Stem Cell Technologies, 15061) from healthy donor leukopaks obtained through the Massachusetts General Hospital Blood Bank, and were cultured in RPMI 1640 media supplemented with 10% fetal bovine serum, 20 U ml$^{-1}$ recombinant human IL-2 (PeproTech), and 1x penicillin/streptomycin at 37 °C in 5% $CO_2$. The non-immortalized PDX PDAC cell lines, PDX 1294, PDX1275, and PDX1319, were kindly provided by the Liss laboratory at Massachusetts General Hospital (MGH) and cultured in DMEM/F12 medium supplemented with 10% fetal bovine serum and 1x penicillin/streptomycin.

### In vivo models

Male and female 6–11 week-old in-house bred NOD.Cg-Prkdcscid Il2rgtm1Wjl/SzJ (NSG) mice (The Jackson Laboratory) were housed in groups of up to five under pathogen-free conditions. The animals were kept at temperatures of 21.1–24.5 °C (70–76 °F), 30–70% humidity, and a 12:12 light-dark cycle. All mice were housed at the MGH Center for Cancer Research, and all care and experiments were conducted in

accordance with protocols approved by the Massachusetts General Hospital Institutional Animal Care and Use Committee (2020N000114). The maximal tumor size permitted (20 mm in diameter for a single tumor or 10 mm for two tumors) was not exceeded in any experiment. All animal procedures, including injections and monitoring, were carried out by three animal technicians who were independent of the study hypotheses and blinded to treatment groups. Both male and female mice were used, and animals were randomized prior to treatment to ensure comparable tumor burden (based on mean caliper measurements) across all groups. Engineered Capan-2 cell lines or PDX cells were resuspended to $5 \times 10^6$ cells per 100 μL of a 1:1 mixture of PBS and Matrigel (Corning) and injected into each mouse subcutaneously. Caliper measurements were taken weekly. Fourteen days after tumor injection, CAR-T cells or untransduced T cells were administered intravenously through the tail vein with the indicated concentration in 100 μL PBS. For the IHC study, mice were euthanized at the indicated time points to assess T cell infiltration by immunohistochemistry. Mouse experiments continued until IACUC guidelines recommended euthanasia. Tumor volume was calculated using the standard formula:

$$\text{Volume} = \left(\text{length} \times \text{width}^2\right)/2$$

### Generation of engineered cell lines
Cell lines were transduced to express click beetle green (CBG) luciferase and enhanced GFP (eGFP), then sorted using a BD FACSAria to isolate a clonal or different population of transduced cells. Capan-2 *C1GALT1* KO was generated by electroporation of 10 μg of Cas9 mRNA and 0.3 nmol of sgRNA (Synthego CRISPRevolution GUAAAGCAGGGCUACAUGAG[17]) using BTX ECM 830 electroporator. Knockout efficiency was measured by flow cytometry, and a pure population was isolated through cell sorting.

### Generation of CAR constructs
Three anti-mesothelin CAR plus bridge constructs (anti-Tn-MUC1, anti-CD19, HPAx2) and nine CAR only constructs (anti-mesothelin, HPAx1, HPAx2, HPAx3, anti-Tn-MUC1, MGLx1, MGLx2, Siglec-15x1, and Siglec-15x2) were synthesized and cloned into a third-generation lentiviral plasmid backbone regulated by a human EF-1α promoter (Genscript). All CAR constructs contained a CD8 hinge and transmembrane domains, a 4-1BB co-stimulatory domain, and an intracellular CD3ζ signaling domain. Bridges were designed to be flanked by an Igκ leader peptide, a CD28 transmembrane domain and mutated CD28 co-stimulatory domain, and an ALFA-tag element. Anti-HPA CAR constructs contained a transgene coding for the fluorescent reporter, mCherry, to aid in evaluating transduction efficiency. The scFvs against mesothelin, Tn-MUC1, and CD19 were derived from sequences of SS1, 5E5, and blinatumomab, respectively (available to the public). Glycobridge constructs were further modified to remove the anti-mesothelin scFv (SS1) or to substitute the CD28 transmembrane domain with the PDGFRβ transmembrane domain using the Q5 Site-Directed Mutagenesis Kit (E0554S; New England Biolabs) or Q5 High-Fidelity 2x Mater Mix (M0492S; New England Biolabs). The sequences of primers and gBlocks are listed in Supplementary Table 1.

### CAR T cell production
Leukapheresis product from anonymous healthy human donors was purchased from the MGH blood bank under an institutional review board–exempt protocol. Donor recruitment, screening, and written informed consent were conducted by the MGH Blood Bank. The investigators received only de-identified samples and had no access to donor information. Stem Cell Technologies T cell Rosette Sep Isolation kit was used to isolate T cells. Bulk human T cells were activated on Day 0 using CD3/CD28 Dynabeads (Life Technologies) at a 1:3 T cell:bead ratio to generate CAR T cells and untransduced T cells from the same donors to serve as controls. T cells were grown in RPMI 1640 media with GlutaMAX and HEPES supplemented with 10% FBS, penicillin, streptomycin, and recombinant human IL-2 (20 IU per ml; Peprotech). On Day 1 (24 h after activation), cells were transduced with CAR lentivirus at an MOI of 5–10. CAR-T cells were expanded with IL-2 containing cell culture media addition every 2–3 days to main the concentration between 0.5 and $1 \times 10^6$ cells/mL. Dynabeads were removed via magnetic separation on Day 6, and cells were assessed by flow cytometry with mCherry expression, ALFA-tag, or anti-$(G_4S)_3$ linker antibody binding on Days 12–14 to determine transduction efficiency prior to cryopreservation. Prior to use in in vitro and in vivo functional assays, CAR T cells and untransduced cells were thawed and rested for 18–24 h in the presence of IL-2.

### Flow cytometric analysis
The following antibody clones were used for CAR-T cell analysis: Alexa Fluor 647 conjugated FluoTag-X2 anti-ALFA (N1502-AF647-L; NanoTag Biotechnologies; 1:100), Alexa Fluor 647 conjugated G4S (E7O2V) antibody (69782S; Cell Signaling Technology; 1:100), Alexa Fluor 700 conjugated anti-human CD3 antibody (300424; BioLegend; 1:100), Per/Cy7 conjugated anti-human CD4 antibody (300518; BioLegend; 1:100), PerCP conjugated anti-human CD8α antibody (301032; BioLegend; 1:100), FITC conjugated anti-human CCR7 antibody (561271; BD BioSciences; 1:50), Brilliant Violet 421 conjugated anti-human CD45RA antibody (304130; BioLegend; 1:100), Alexa Fluor 700 conjugated anti-human CD45 antibody (304024; BioLegend; 1:100). Each antibody was diluted in PBS containing 2% FBS, as indicated. Adherent cancer cells were detached by incubating with Tryple Express Enzyme (1x; 12-604-013; Fisher Scientific) at 37 °C for 5–10 min. Alexa Fluor 647 conjugated HPA lectin (L32454; Thermo Scientific), PE conjugated CD19 (363003; BioLegend), and APC conjugated anti-human CD69 (310910; BioLegend) were diluted 1:200 in 2% FBS PBS and incubated with cells at 4 °C for 1 h for each stain. For analysis of Tn-MUC1 cell surface expression level, anti-human Tn-MUC1 (FHD14210-100; ProteoGenix) was diluted 1:200 in 2% FBS PBS and incubated with cells at 4 °C for 1 h. Secondary labeling was with Alexa Fluor 647 conjugated goat anti-human IgG (H + L) recombinant secondary antibody (A56019; Invitrogen), diluted 1:200 in 2% FBS PBS and incubated with cells at 4 °C for 1 h. To quantify cell surface CD19 expression level, BD Quantibrite Beads PE Fluorescence Quantification Kit (340495; BD Biosciences) was used according to the manufacturer's protocol. Cells were washed and stained with DAPI containing 2% FBS PBS to assess cell viability before analyzing on a BD Fortessa X-20.

### Cytotoxicity assays
For single time-point cytotoxicity assays, target cells expressing CBG luciferase were incubated with CAR-T cells or untransduced T cells at varying effector-to-target (E:T) ratios in 200 μL of growth media specific to the target cell line, without IL-2, and cocultured in a 96-well plate for 30 h at 37 °C in 5% $CO_2$. Cells were then lysed using the Bright-Glo Luciferase Assay System (E2610; Promega), and luciferase activity was measured with a Biotek Neo2 luminescence plate reader. Specific lysis was calculated using the following formula: Percentage of specific lysis = [(luminescence target cell only) – (luminescence target cell + CAR-T cell)]/(luminescence target cell only) × 100%. For real-time killing assays, target cells expressing CBG-eGFP were plated in flat-bottom 48- or 96-well plates (Corning) and incubated for at least 4 h at 37 °C in 5% $CO_2$. CAR-T cells or untransduced T cells were added in triplicate at various effector:target (E:T) ratios as indicated. CAR expression (%) was normalized using untransduced T cells for each donor. Plates were then incubated at 37 °C for up to 6 days, with whole wells recorded every 2 h using the IncuCyte Live Cell Analysis system. Cytotoxicity was measured as the total green fluorescent area and analyzed using the IncuCyte image analysis software.

## Immunofluorescence

Target cancer cell lines (Capan-2, Capan-2 *C1GALT1* KO, and PDX1294) were plated in 35 mm glass bottom dishes (P35G-1.5-14-C; Mattek), grown for 24 h. HPA conjugated Alexa Fluor 647 (L32454; Thermo Scientific) was diluted 1:200 in 2% FBS PBS and incubated on samples at 4 °C for 1 h. For itraconazole treatment, Capan-2 cells were treated with 2.5 μM itraconazole in cell culture media at 37 °C for 48 h before staining with HPA lectin. Cells were further treated with Hoechst 33342 (4082S; Cell Signaling Technology) containing 2% FBS PBS. All samples above were imaged on a LSM 780 confocal microscope using 63× (NA: 1.4 Oil) objectives (Zeiss).

## Immunohistochemistry (IHC)

After tumor-engrafted mice were euthanized, tumors were extracted and fixed in 4% PFA overnight, washed with PBS, and serial wash with 30, 50, and 70% ethanol and stored in 70% EtOH until staining. Tissue slides were then made and embedded in paraffin. Slides were stained for CD3, Tn antigen (SBH-Tn-100ug, SBH Sciences), and HPA-biotin (L6512, Sigma-Aldrich) by the specialized histopathology services core facility at MGH. All IHC slides were imaged on a Axio Scan.Z1 microscope and quantified the respective stains by QuPath (v0.5.1).

## Cell avidity measurement with acoustic force microscopy

Capan2, Capan-2 *C1GALT1* KO, and PDX1294 cells were seeded in Poly-L Lysine-coated z-Movi chips (Lumicks) at a density of $60 \times 10^6$ cells/mL and incubated for 2 h. CAR-T cells were sorted using a Sony MAB900 (Sony Biotechnology Inc.) 48 h before the avidity assessment, with anti-$(G_4S)_3$ linker conjugated with Alexa Fluor 647 and mCherry$^+$ gates to isolate only CAR + T cells. Sorted CAR-T cells and untransduced T cells were stained with a CellTrace™ Far Red Proliferation Kit (Thermo Fisher Scientific). Each chip was first run with untransduced T cells to prevent nonspecific confounding binding of CAR-T cells, followed by CAR-T cells with three different bridges or HPA-based CAR, and finally untransduced T cells. To eliminate any potential bias in binding from earlier runs, the order of CAR-T cells to be tested was alternated. T cells were incubated for a 5-min binding period and visualized on the z-Movi Cell Avidity Analyzer (Lumicks), while a gradient acoustic force of up to 1000 pN was applied. The percentage of cells bound as a function of the acoustic force applied was then analyzed using Ocean software (version 1.5.5, Lumicks). Each chip was run five times.

## Statistical methods, sample sizes, data collection, and assumptions

The sample sizes were selected on the basis of standards in the field and not pre-determined using statistical methods. For statistical comparisons, data distributions were assumed to be normal. Normality was tested for conditions with approximately ten or more data points. All the statistical analyses were performed using GraphPad Prism 8 software. All the experimental data are presented as mean ± s.d. or as box-and-whisker plots with the first and third quartiles (boxes), median and range of data, unless stated otherwise within figure legends. Appropriate statistical tests were used to analyze the data, as described in the figure legends.

## Reporting summary

Further information on research design is available in the Nature Portfolio Reporting Summary linked to this article.

## Data availability

All data shown in this manuscript are provided in the Article, Supplementary Information and Source data files. Source data are provided with this paper. All additional data and materials that can be shared will be released using a material transfer agreement. Please contact M.V.M. at mvmaus@mgh.harvard.edu. Source data are provided with this paper.

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

## Acknowledgements

We thank the core facilities at the MGH Cancer Center: Flow Cytometry, Histopathology, and Blood Bank. S.P. is a Merck Fellow of the Damon Runyon Cancer Research Foundation (DRG-2529-24). D.S.B. received funding from the CRIS Foundation Out-Back Fellowship Programme (outback2021_6) and from the Spanish Society of Medical Oncology (SEOM). F.B. received funding from the American-Italian Cancer Foundation (AICF) and the Italian Association for Cancer Research (AIRC). G.E. received funding from the Gilead Scholars Program in Oncology Solid Tumors. M.B.L. is supported by NIH K08 (K08CA289948). This work was funded by NIH R01 CA238268 (M.V.M.) and NIH K99CA303346 (S.P.)

## Author contributions

S.P. and M.V.M. designed the research. S.P., C.H., and A.W. performed experiments. E.D. and H.T. isolated and validated Jeko-1 clones. All authors (S.P., C.H., E.D., A.W., H.T., F.B., A.A.B., D.S.B., G.E., M.B.L., A.M., T.R.B., and M.V.M) made a significant contribution to the discussion. S.P., T.R.B., and M.V.M. wrote the manuscript with feedback from all authors.

## Competing interests

M.V.M. and S.P. are inventors on patent applications filed by MGH related to CAR-T cell technologies described in this study. All other authors declare no competing interests. M.V.M. is an inventor on patents related to adoptive cell therapies, held by Massachusetts General Hospital (some licensed to Promab and Luminary) and University of Pennsylvania (some licensed to Novartis). M.V.M. holds equity in 2SeventyBio, A2Bio, Affyimmune, BendBio, Cargo, GBM newco, Model T bio, Neximmune, and Oncternal. M.V.M. receives Grant/Research support from Kite Pharma, Moderna, and Sobi. M.V.M. has served as a consultant for multiple companies involved in cell therapies. M.V.M.'s competing interests are managed by Mass General Brigham. M.B.L. is a contributor to patent filings on CAR-T technology held by the Massachusetts General Hospital and has served as a consultant for BioNtech, Cabaletta Bio, and Onclive.
