## [Transparent Peer Review file · Nature Communications]

Tuning CAR-T cells by targeting cancer-associated glycan in pancreatic cancer

Corresponding Author: Dr Marcela Maus

Version 0:

Reviewer comments:

Reviewer #1

(Remarks to the Author)

The authors have significantly revised the manuscript. All my questions raised have been addressed within the revised manuscript.

Reviewer #2

(Remarks to the Author)

I particularly appreciated how thoroughly the authors responded to all the questions, providing detailed explanations that reflect a genuine effort to understand the reviewers' perspectives and improve the manuscript.

The authors have addressed both major and minor comments and revised the manuscript accordingly, incorporating new experimental data that enhance its clarity and impact. These additions include:

- i) Cytotoxicity assays demonstrating the activity of CAR-T cells equipped with either HPA-bridge or Tn-MUC1-bridge modules against multiple patient-derived xenograft (PDX) models;
 - ii) HPA lectin staining on organs from mice bearing Capan-2 C1GALT1 knockout tumors and
 - iii) HPA-biotin staining on a human pancreatic tissue microarray, encompassing both tumor and adjacent normal tissues.
- These additions have improved the clarity of both the text and the figures, and the manuscript has clearly benefited from the revision process.

However, I think the study still lacks a strong conceptual focus. It remains unclear whether its primary contribution lies in technological innovation, mechanistic insight, or translational relevance. Throughout the point-by-point reply, the authors describe the study as a "proof-of-concept". Proof-of-concept studies published in high-impact journals should ideally present disruptive innovations. In this case, the advantage appears more incremental, as the strategy essentially represents a novel dual-antigen CAR-T design that uses aberrantly glycosylated mucins as accessory targets, without a deep exploration or exploitation of glycobiology concepts (despite the introduction being strongly focused on this point).

I think that this does not necessarily diminish the relevance of the work, provided that the manuscript shifts toward a stronger translational focus, particularly in terms of safety. Although the toxic profile observed in the murine model (Fig. 5) is encouraging, its translational relevance depends on whether antigen distribution in mice reflects that in humans. Supporting data or just references confirming this similarity would strengthen the safety argument. In the absence of such evidence, it becomes even more important to assess target expression directly in human tissues expressing the CAR targets (mesothelin).

Reviewer #3

(Remarks to the Author)

The authors have made a commendable effort to revise the manuscript in response to the prior review with additional data

and commentary to clarify important issues for the reader. The noteworthy result is that improving adhesion, in this case by incorporating binding domains that target the glycocalyx but do not signal can improve CAR T cell activity against solid tumors that overexpress Tn antigens. The importance of increasing avidity by incorporating a second binder (signaling or not) has been previously shown, although not referred to as a "bridge". Thus, the novelty in this paper is incremental by showing that glycosylated molecules are suitable targets for increasing adhesion. This is perhaps not surprising however the experiments are carefully performed, the conclusions are well supported, and the discussion acknowledges other work with alternative target molecules. The authors also acknowledge that several of the additional experiments suggested by the prior review to further pursue mechanisms are important but argue they are beyond the scope of this "proof of principle" manuscript. Given the extensive data already presented, I would be inclined to accept this argument. There are a couple of issues that would improve the paper:

1. The authors suggest in their introduction that "designing CAR T cells that penetrate the glycolcalyx...." This is a vast overstatement as there is no experimental evidence presented that the T cells actually penetrate the glycocalyx -- it is more likely that, as supported by the data, their approach sustains adhesion and enables better CAR T cell activation.
2. There is substantial redundancy in the introduction and the results, the writing could be more concise.

Reviewer #4

(Remarks to the Author)

This manuscript has been previously reviewed and the authors have responded with great attention to the initial review. There are new experiments performed that increase the rigor of the work, and better address the potential off target effects. Importantly, in this revision, they have taken appropriate steps to clarify the specific innovation within the report when considered in the context of the larger field of data. Overall, the report has some innovative features in refining CAR T cell design in models of pancreatic cancer, which may be adaptable to other settings.

There are no further suggestions for this manuscript and it seems a high level of effort has been given to address the initial review.

Point-by-point response to reviewer comments

We thank the reviewers for their time and excellent feedback. In response to the reviews, we have revised the manuscript with edits to the text. Our point-by-point responses appear below.

Reviewer #1 (Remarks to the Author):

The authors have significantly revised the manuscript. All my questions raised have been addressed within the revised manuscript.

Reviewer #2 (Remarks to the Author):

I particularly appreciated how thoroughly the authors responded to all the questions, providing detailed explanations that reflect a genuine effort to understand the reviewers' perspectives and improve the manuscript.

The authors have addressed both major and minor comments and revised the manuscript accordingly, incorporating new experimental data that enhance its clarity and impact. These additions include:

- i) Cytotoxicity assays demonstrating the activity of CAR-T cells equipped with either HPA-bridge or Tn-MUC1-bridge modules against multiple patient-derived xenograft (PDX) models;
- ii) HPA lectin staining on organs from mice bearing Capan-2 C1GALT1 knockout tumors and
- iii) HPA-biotin staining on a human pancreatic tissue microarray, encompassing both tumor and adjacent normal tissues.

These additions have improved the clarity of both the text and the figures, and the manuscript has clearly benefited from the revision process.

However, I think the study still lacks a strong conceptual focus. It remains unclear whether its primary contribution lies in technological innovation, mechanistic insight, or translational relevance. Throughout the point-by-point reply, the authors describe the study as a “proof-of-concept”. Proof-of-concept studies published in high-impact journals should ideally present disruptive innovations. In this case, the advantage appears more incremental, as the strategy essentially represents a novel dual-antigen CAR-T design that uses aberrantly glycosylated mucins as accessory targets, without a deep exploration or exploitation of glycobiology concepts (despite the introduction being strongly focused on this point).

I think that this does not necessarily diminish the relevance of the work, provided that the manuscript shifts toward a stronger translational focus, particularly in terms of safety. Although the toxic profile observed in the murine model (Fig. 5) is encouraging, its translational relevance depends on whether antigen distribution in mice reflects that in humans. Supporting data or just references confirming this similarity would strengthen the safety argument. In the absence of such evidence, it becomes even more important to assess target expression directly in human tissues expressing the CAR targets (mesothelin).

We thank the reviewer for the thoughtful comments and agree with the point regarding the proof-of-concept nature of this work. To address the reviewer’s concern, we have added references demonstrating that mesothelin expression patterns, biochemical characteristics, and tissue distribution in mice are similar to those in humans (Zervos et al., 2016; Hagerty et al., 2021). In addition, our in vivo studies were performed in NSG mice engrafted with human pancreatic cancer cells (Capan-2 and patient-derived xenografts), which express human mesothelin

rather than the murine ortholog. Therefore, the safety and efficacy data presented here directly reflect the interaction between human CAR-T cells and human mesothelin-expressing tumor cells in a controlled in vivo setting. Together, these additions strengthen the rationale for our model choice, clarify the relevance of our murine safety data, and support the discussion of antigen distribution and on-target, off-tumor risk.

These points have been incorporated into the revised discussion with text and references as follows:

“Moreover, given that mesothelin is one of the key CAR targets in pancreatic models, its translational relevance depends on whether antigen distribution in mice reflects that in humans. Prior studies have shown that the expression pattern, biochemical characteristics, and tissue distribution of mesothelin in mice are similar to those in humans.”

Reviewer #3 (Remarks to the Author):

The authors have made a commendable effort to revise the manuscript in response to the prior review with additional data and commentary to clarify important issues for the reader. The noteworthy result is that improving adhesion, in this case by incorporating binding domains that target the glycocalyx but do not signal can improve CAR T cell activity against solid tumors that overexpress Tn antigens. The importance of increasing avidity by incorporating a second binder (signaling or not) has been previously shown, although not referred to as a "bridge". Thus, the novelty in this paper is incremental by showing that glycosylated molecules are suitable targets for increasing adhesion. This is perhaps not surprising however the experiments are carefully performed, the conclusions are well supported, and the discussion acknowledges other work with alternative target molecules. The authors also acknowledge that several of the additional experiments suggested by the prior review to further pursue mechanisms are important but argue they are beyond the scope of this "proof of principle" manuscript. Given the extensive data already presented, I would be inclined to accept this argument. There are a couple of issues that would improve the paper:

1. The authors suggest in their introduction that "designing CAR T cells that penetrate the glycolcalyx...." This is a vast overstatement as there is no experimental evidence presented that the T cells actually penetrate the glycocalyx -- it is more likely that, as supported by the data, their approach sustains adhesion and enables better CAR T cell activation.

We thank the reviewer for this thoughtful feedback and catching this out. We agree and have revised the wording to “binds effectively to glycocalyx,” which is consistent with our data showing sustained adhesion and improved activation rather than physical penetration. We have made this change in the Introduction and ensured consistency in the Abstract and Introduction.

This point has been incorporated into the revised introduction with text as follows:

“Overcoming this barrier by designing CAR-T cells that can bind the glycocalyx effectively may significantly enhance their ability to bind their target antigen and subsequently induce tumor cell death, thereby improving therapeutic efficacy in solid cancers like pancreatic cancer.”

2. There is substantial redundancy in the introduction and the results, the writing could be more concise.

We agree with the reviewer's comment and have revised the introduction to eliminate redundancy.

“However, glycoproteins and other large components of the glycocalyx are thought to sterically shield molecular epitopes, affecting the interactions between immune cells and tumor cells. Aberrant N-glycosylation, such as β 1,6-GlcNAc-branched N-glycans synthesized by MGAT5, has been shown to promote immune evasion by masking tumor-associated antigens and contributing to T cell dysfunction and exhaustion. Overexpression of MUC1 contributes to a thickened glycocalyx, preventing access to key receptors and/or antigens needed for immune cell killing.

Our results reveal that glycan-targeting strategies using a bridge system that binds to cancer-associated glycans enhance CAR-T cell-mediated cytotoxicity.”

Reviewer #4 (Remarks to the Author):

This manuscript has been previously reviewed and the authors have responded with great attention to the initial review. There are new experiments performed that increase the rigor of the work, and better address the potential off target effects. Importantly, in this revision, they have taken appropriate steps to clarify the specific innovation within the report when considered in the context of the larger field of data. Overall, the report has some innovative features in refining CAR T cell design in models of pancreatic cancer, which may be adaptable to other settings.

There are no further suggestions for this manuscript and it seems a high level of effort has been given to address the initial review.